# Effect of Vacancy Defect Content on the Interdiffusion of Cubic and Hexagonal SiC/Al Interfaces: A Molecular Dynamics Study

**DOI:** 10.3390/molecules28020744

**Published:** 2023-01-11

**Authors:** Masoud Tahani, Eligiusz Postek, Leili Motevalizadeh, Tomasz Sadowski

**Affiliations:** 1Department of Mechanical Engineering, Ferdowsi University of Mashhad, Mashhad 91779-48974, Iran; 2Institute of Fundamental Technological Research, Polish Academy of Sciences, Pawińskiego 5B, 02-106 Warsaw, Poland; 3Department of Physics, Mashhad Branch, Islamic Azad University, Mashhad 91871-47578, Iran; 4Department of Solid Mechanics, Lublin University of Technology, 20-618 Lublin, Poland

**Keywords:** interdiffusion, diffusion coefficient, SiC/Al interface, vacancy, molecular dynamics

## Abstract

The mechanical properties of ceramic–metal nanocomposites are greatly affected by the equivalent properties of the interface of materials. In this study, the effect of vacancy in SiC on the interdiffusion of SiC/Al interfaces is investigated using the molecular dynamics method. The SiC reinforcements exist in the whisker and particulate forms. To this end, cubic and hexagonal SiC lattice polytypes with the Si- and C-terminated interfaces with Al are considered as two samples of metal matrix nanocomposites. The average main and cross-interdiffusion coefficients are determined using a single diffusion couple for each system. The interdiffusion coefficients of the defective SiC/Al are compared with the defect-free SiC/Al system. The effects of temperature, annealing time, and vacancy on the self- and interdiffusion coefficients are investigated. It is found that the interdiffusion of Al in SiC increases with the increase in temperature, annealing time, and vacancy.

## 1. Introduction

Silicon carbide (SiC) ceramics have electronic applications such as LEDs and detectors and are used in semiconductor devices that work at high temperatures and voltages. SiC has excellent properties, including its low density, high specific strength, low thermal expansion and conductivity, high thermal stability, outstanding wear resistance, and great corrosion resistance, etc. [1]. With these unique properties, SiC can be used in a wide range of applications, such as cutting tools, gas turbines, and in the aerospace, automobile, and chemical industries. However, some of the disadvantages of SiC are its brittleness, relatively low thermal conductivity, low fracture toughness and strength, and poor resistance to creep, fatigue, and thermal shock. The disadvantages of ceramics can be overcome by microstructural engineering in the development of metal matrix composites (MMCs).

The SiC/Al is an MMC consisting of silicon carbide reinforcements dispersed in a matrix of aluminum alloy. It combines the benefits of the unique properties of silicon carbide and the good mechanical properties of aluminum. The SiC/Al MMCs have extensive applications in microelectronic packaging for aerospace, automotive, and microwave applications due to their excellent properties, such as their low density, high strength, high toughness, high fatigue strength, excellent mechanical damping, low coefficient of thermal expansion, good wear resistance, and so on [2,3].

In the small-scale production of SiC, a variety of defects occurs. The carbon and silicon vacancy defects have been observed as the most important point defects in SiC [4,5,6,7]. For example, Kukushkin and Osipov [7] studied the mechanism of the formation of carbon vacancy in SiC during its growth by atomic substitution. Furthermore, Janzén et al. [5] presented an experimental method to specify the levels of silicon vacancies in 4H- and 6H-SiC.

Metal matrix composites usually consist of a few phases and some additions; therefore, besides the properties of each phase, the interface properties severely affect the overall properties of the nanocomposite. Postek and Sadowski [8,9], for example, showed the significance of the interface properties between the phases in WC/CO composites by employing cohesive law between phases. The interface between grains and phases at the atomic level is indeed a mixture of atoms of phases connected because of diffusion. To this end, the equivalent material properties of the interface between phases play an essential role in the mechanical response of MMCs to loading. A cohesive zone model with the traction–separation law is traditionally utilized to characterize the interface. The traction–separation relationships are determined by performing mode I and mode II failure tests for a ductile–brittle system.

Moya et al. [10] investigated the challenges of ceramic/metal micro/nanocomposites in the new technologies. They reviewed the exotic effects of metal particles embedded into matrix ceramics due to the dissimilar properties of the components, percolation laws, and the nature of the interfaces. The interested reader will find sufficient references in this review article to cover the literature in more depth concerning several aspects of ceramic/metal interfaces, including the role of the interface in the fracture toughness and wear resistance of the composite and the wettability issue of dissimilar ceramic and metal materials to reach an appropriate adherence.

The diffusion in solids occurs due to the thermally activated random motion of atoms. Interdiffusion or diffusion coupling is a process of diffusional exchange of atoms across two materials that are in contact. The diffusion of Al into SiC at temperatures between 1700 to 2400 °C is investigated by Chang et al. [11]. Mokhov et al. [12] determined the diffusion constant of Al containing vapor in SiC at different temperatures using experimental data. Van Opdorp [13] showed that the penetration depth of Al into SiC from the vapor source was more considerable than that from the solid source. Tajima et al. [14] studied the diffusion constant of Al into SiC at a temperature between 1350 to 1800 °C and characterized a low activation energy and low pre-exponential constant compared with previously reported results. They observed that self- and most impurity diffusion in SiC occurs by a vacancy.

Tham et al. [15] synthesized SiC/Al composites using the melt deposition technique by pre-heating the SiC particles in the air for 60 min at 950 °C and mechanically stirring the fully molten aluminum alloy superheated to 950 °C. They observed the formation of a thin Al_4_C_3_ reaction layer along the particle–matrix interface. Lee et al. [16] stated that the formation of brittle and unstable Al_4_C_3_, according to the reaction $4\mathrm{Al}+3\mathrm{SiC}\;\to {\;\mathrm{Al}}_{4}C_{3}+3\mathrm{Si}$, at the SiC/Al interface degrades the mechanical properties of the composite, and, hence, its formation during composite fabrication must be either avoided or minimized. Lee et al. [16] suggested the addition of Si into the matrix, coating of SiC, and the passive oxidation of SiC to avoid the formation of Al_4_C_3_ and to obtain the desired SiC/Al interface. Sozhamannan and Prabu [17] produced samples with interface bonding of SiC/Al at various processing temperatures, and evaluated the interface compounds using an energy dispersive spectroscope. They measured the diffusion length and estimated the interface characteristics by tensile and microhardness tests. Soloviev et al. [18] investigated the diffusion of Al in 4H-SiC substrates with different orientations at temperatures of 1900 to 2000 °C, and measured their impurity profiles using secondary ion mass spectrometry (SIMS). The diffusion of Al in 4H-SiC during postimplantation annealing was studied by Müting et al. [19], who found that Al diffuses with a low diffusion rate in SiC during the heat treatment using defect-enhanced diffusion mechanisms. Tahani et al. [20] investigated the interdiffusion of Al in 6H-SiC and 3C-SiC at temperatures of 1000 to 2000 K using molecular dynamics. They found that the Si-terminated interface in the 6H-SiC/Al diffusion couple has a higher diffusivity than the C-terminated one, while the opposite is true for the 3C-SiC/Al diffusion couple.

In the present study, the self-diffusion and interdiffusion at the interface of SiC/Al with vacancy defects in SiC are investigated using the molecular dynamics (MD) method. The C- and Si-terminated interfaces of α-SiC particulate-reinforced Al and β-SiC whisker-reinforced Al composites are considered. The average ternary interdiffusion coefficients are evaluated by the method proposed by Dayananda and Sohn [21]. The interested reader will find sufficient references on methods for the determination of the interdiffusion coefficients in metallic solids in the review article by Zhong et al. [22]. The effects of temperature, annealing time, and vacancy on the self-diffusion and main and cross-interdiffusion coefficients of components are studied.

## 2. Results and Discussions

### 2.1. Self-Diffusion

The diffusion characteristics of the SiC/Al interface are studied by heating the system to a preset temperature and maintaining it for 6.0 ns at this temperature. Figure 1 illustrates the evolution of the interface diffusion for different snapshots with 2.0 ns intervals for the C-terminated 6H-SiC/Al and 3C-SiC/Al interfaces with 20% vacancy in SiC and heating up to 1000 K. Figure 1a,f shows the sharp interface between SiC and Al considering an initial gap equal to the C–Al bond length. It can be seen from Figure 1b,g that, before the temperature reaches 1000 K, the Al atoms move locally near the interface because of a strong interfacial bond between Al and SiC. Hoekstra and Kohyama [23], using the ab initio pseudopotential method, showed that the C–Al bond is almost twice as strong as the Si–Al bond, and, in general, the interfacial bond between the SiC and Al is stronger than the intralayer bonds within the pure aluminum. By maintaining the system at 1000 K, more atoms pass through the interface, and a thicker fuzzy interface is produced. The diffusion zones after maintaining the systems at 1000 K for 6 ns are illustrated in Figure 1e,j.

To illustrate clearly the diffusion of Al atoms in the SiC with 20% vacancy, Figure 2 shows the front and top views of the configuration of Al atoms in the diffusion zone of the C-terminated 6H-SiC/Al and 3C-SiC/Al after maintaining the systems at 1000 K for 0 ns to 6 ns. It can be seen that, as the maintaining time of the system in the annealing temperature increases, the number of atoms crossing the interface also increases, and the thickness of the diffusion zone becomes thicker.

Figure 3 shows the concentration profiles of Al, Si, and C atoms along the *z*-direction, perpendicular to the interface plane, for the C-terminated 6H-SiC/Al and 3C-SiC/Al interfaces with 20% vacancy in SiC at some selected times. Each system is cut into thin slices of thickness 4 Å along the *z*-direction, and the number of each atom type is counted to obtain the concentration. Figure 3a,f illustrates the initial concentration profiles before diffusion. Next, as the systems are heated up to 1000 K, the diffusion begins, and, as expected, the thickness of the diffusion zone increases. The diffusion zone is marked with grey color in each figure. As is seen, the thickness of the diffusion zone increases rapidly by maintaining the systems at 1000 K for 2 ns, and then the speed of the increase in the thickness of the diffusion zone decreases. The thickness of the diffusion zone after maintaining the systems at 1000 K for 6 ns reaches 31 Å in 6H-SiC/Al and 32 Å in 3C-SiC/Al.

The self-diffusion coefficients of each atom are determined from the slope of the mean square displacements (MSDs) using Einstein’s relation Equation (1) [24]:(1)$$D_{A}=\underset{t\to \infty }{\lim}\frac{1}{N_{A}}\sum_{i=1}^{N_{A}}\frac{\langle {\left|r_{i}^{A}(t)-r_{i}^{A}(0)\right|}^{2}\rangle }{6t}$$
where *N_A_* is the number of atoms *A*, and $r_{i}^{A}$ is the position vector of the *i*th atom of type *A*. In Equation (1), 〈⋯〉 denotes the average over all atoms of the same type. Figure 4 shows the time history of the self-diffusion coefficients of Al, Si, and C atoms in the defective C- and Si-terminated 6H-SiC/Al and 3C-SiC/Al interfaces with 20% vacancy in SiC for various temperatures. The time in this figure is started when the systems are heated up from 300 K to the preset temperature with a heating rate up of 1 K/ps. It is observed that the self-diffusion coefficients of Al atoms in the C-terminated systems are slightly larger than those of the Si-terminated ones; however, for Si and C atoms, they are vice versa. Furthermore, as expected, the maximum self-diffusion coefficients of Al atoms are almost 200 times higher than those of Si and C atoms at the same temperature. To obtain the activation energy *Q* and pre-exponential factor *D*_0_ of atoms, the Arrhenius equation $D=D_{0}\exp\left(-Q/RT\right)$ is fitted to the data of Figure 5 and the same data not shown here for the defective systems with 10% vacancy in SiC. The Arrhenius plots of Al, Si, and C atoms for the C- and Si-terminated 6H-SiC/Al and 3C-SiC/Al interfaces with 10% and 20% vacancies in SiC are illustrated in Figure 5. The results of the activation energies and pre-exponential factors of atoms are also presented in Table 1. It is worth mentioning that, in the Arrhenius plot of Al atoms, the annealing temperature of 1000 K is not considered, since it is above the melting temperature of Al (i.e., 933 K).

### 2.2. Interdiffusion

The interdiffusion flux ${\tilde{J}}_{i}$ of component *i* and its concentration gradient $\partial C_{i}/\partial z$ for a binary system in an isothermal condition is given by Fick’s law. In general, the interdiffusion of a multi-component system containing *n* components can be expressed by Onsager’s formalism [25,26] of Fick’s law (Equation (2)):(2)$${\tilde{J}}_{i}=-\sum_{j=1}^{n-1}{\tilde{D}}_{ij}^{n}\frac{\partial C_{j}}{\partial z}$$
where ${\tilde{D}}_{ij}^{n}$ is the interdiffusion coefficients, and *C_i_* is the mole fraction of component *i*. In total, four independent interdiffusion coefficients, ${\bar{\tilde{D}}}_{11}^{3},\;\;{\bar{\tilde{D}}}_{12}^{3},\;\;{\bar{\tilde{D}}}_{21}^{3},$ and ${\bar{\tilde{D}}}_{22}^{3}$, are required to describe the present ternary diffusion system, and they can be determined by using the Boltzmann–Matano [27,28] method. The interdiffusion fluxes for each component *i* at time *t* can be determined from the concentration profiles as (Equation (3)) [29]:(3)$${\tilde{J}}_{i}(z)=\frac{1}{2t}\int_{C_{i}^{-\infty }\;\mathrm{or}\;C_{i}^{+\infty }}^{C_{i}^{z}}(z-z_{0})dC_{i}$$
where $C_{i}^{-\infty }$ and $C_{i}^{+\infty }$ are the mole fraction of component *i* at the bottom and top terminal ends of the diffusion couple, respectively, and *z*_0_ is the position of the Matano plane, which, by assuming $z_{}^{-\infty }=0$, is obtained by (Equation (4)):(4)$$z_{0}=\frac{1}{C_{i}^{+\infty }-C_{i}^{-\infty }}\int_{C_{i}^{-\infty }}^{C_{i}^{+\infty }}zdC_{i}$$

In the present study, the method proposed by Dayananda and Sohn [21] is utilized to obtain the average ternary interdiffusion coefficients. This method uses only a single diffusion couple, and the average values of the obtained main and cross-interdiffusion coefficients are consistent with those determined by the Boltzmann–Matano analysis in the diffusion zone.

The four equations are obtained by applying the following equations to any two components in a ternary system (Equations (5) and (6)) [21]:(5)$$\int_{z_{1}}^{z_{2}}{\tilde{J}}_{i}dz={\bar{\tilde{D}}}_{i1}^{3}\left(C_{1}^{z_{1}}-C_{1}^{z_{2}}\right)+{\bar{\tilde{D}}}_{i2}^{3}\left(C_{2}^{z_{1}}-C_{2}^{z_{2}}\right)$$
(6)$$\int_{z_{1}}^{z_{2}}{\tilde{J}}_{i}\left(z-z_{0}\right)dz=2t\left\{{\bar{\tilde{D}}}_{i1}^{3}[{\tilde{J}}_{1}(z_{1})-{\tilde{J}}_{1}(z_{2})]+{\bar{\tilde{D}}}_{i2}^{3}[{\tilde{J}}_{2}(z_{1})-{\tilde{J}}_{2}(z_{2})]\right\}$$

By solving these four algebraic equations, four average ternary interdiffusion coefficients (main and cross), ${\bar{\tilde{D}}}_{11}^{3},\;\;{\bar{\tilde{D}}}_{12}^{3},\;\;{\bar{\tilde{D}}}_{21}^{3},$ and ${\bar{\tilde{D}}}_{22}^{3}$, are determined, which are treated as constants over the compositional range. The four equations are independent and have unique solutions by selecting nonlinear segments of the profiles ${\tilde{J}}_{i}$ and ${\tilde{J}}_{i}(z-z_{0})$ [21]. The criteria for validating the calculated ternary interdiffusion coefficients for each of the couples are as follows (Equation (7)) [30]:(7)$${\bar{\tilde{D}}}_{11}^{3}+{\bar{\tilde{D}}}_{22}^{3}>0\;{\left({\bar{\tilde{D}}}_{11}^{3}+{\bar{\tilde{D}}}_{22}^{3}\right)}^{2}\geq 4\left({\bar{\tilde{D}}}_{11}^{3}{\bar{\tilde{D}}}_{22}^{3}-{\bar{\tilde{D}}}_{12}^{3}{\bar{\tilde{D}}}_{21}^{3}\right)\;\left({\bar{\tilde{D}}}_{11}^{3}{\bar{\tilde{D}}}_{22}^{3}-{\bar{\tilde{D}}}_{12}^{3}{\bar{\tilde{D}}}_{21}^{3}\right)>0$$

The obtained interdiffusion coefficients should satisfy these criteria to ensure the validity of the procedure and computations.

To fit the concentration curve for each component, the Gaussian error function and Boltzman function [31] can be used, and both yield almost the same results. However, in the present study, the Gaussian error function for each component *i* is used, which can be written as follows (Equation (8)):(8)$$C_{i}(t,z)=p_{1i}+p_{2i}erf\left(\frac{z-p_{3i}}{p_{4i}}\right)$$
where *p*_1*i*_ to *p*_4*i*_ are the fit parameters to be determined for each component. The curve fit is performed, and the coefficients are obtained in each time step. The fitted curves are employed in Equations (5) and (6) for two components of the present ternary system to obtain the four independent interdiffusion constants.

In this section, the systems are minimized using the conjugate gradient algorithm, followed by the NVT and NPT ensembles, as described in Section 3, to relax the systems. Thereafter, the systems are maintained at the given temperature for 6 ns to study the interdiffusion. The profiles of the interdiffusion fluxes $\tilde{J}$ and $\tilde{J}(z-z_{0})$ for the diffusion systems of the C-terminated 6H-SiC/Al and 3C-SiC/Al with 10% and 20% vacancy defects in SiC after maintaining the systems at 1000 K for 6 ns are shown in Figure 6. The Matano plane obtained from Equation (4) is marked with a dashed vertical line and *z*_0_ in this figure. The Al and Si atoms are arbitrarily chosen as independent variables, and the C atom is selected as the dependent variable. It can be seen that the maximum interdiffusion fluxes of Al and Si atoms in 6H-SiC/Al increase by 40.6% and 35.6%, respectively, as the vacancy in SiC increases from 10% to 20%. Similarly, the increase in the maximum interdiffusion fluxes of Al and Si atoms for 3C-SiC/Al is 52.4% and 62.5%, respectively. Moreover, for systems with 20% vacancy in SiC, the maximum interdiffusion fluxes of Al and Si atoms in 6H-SiC/Al are 5.9% and 8.1% smaller than those of 3C-SiC/Al, respectively.

The average values of the main and cross-interdiffusion coefficients of the C- and Si-terminated 6H-SiC/Al and 3C-SiC/Al systems with 10% and 20% vacancy defects in SiC are presented in Table 2 and Table 3, respectively. The diffusion systems are maintained for 6 ns at an annealing temperatures of 700 to 1000 K. The interdiffusion coefficients are obtained for composition ranges at the bottom and top sides of the Matano plane. It is observed that the main interdiffusion coefficients for the bottom and top sides of the Matano plane are the same up to three decimal places. Furthermore, all cross-interdiffusion coefficients are at least four orders of magnitude smaller than the main interdiffusion coefficients. As expected, by increasing the annealing temperature, the main interdiffusion coefficients also increase. Moreover, the results in these tables indicate that the diffusivity of Si atoms (i.e., ${\bar{\tilde{D}}}_{22}^{3}$) is more for the Si-terminated systems compared with the C-terminated ones. It is worth mentioning that the average values of the interdiffusion coefficients ${\bar{\tilde{D}}}_{ij}^{3}$ approach the true values ${\tilde{D}}_{ij}^{3}$ if the interval between *z*_1_ and *z*_2_ is sufficiently small [32].

To compare the interdiffusion coefficients of the defective diffusion systems with those of the defect-free ones, the interdiffusion coefficients of the intact form of the aforementioned systems for the annealing temperature of 1000 K and the annealing time of 6 ns are also tabulated in Table 4. It can be seen that, as expected, the interdiffusion coefficients increase rapidly as the vacancy defect increases.

Figure 7 shows the variations in the average main interdiffusion coefficients ${\bar{\tilde{D}}}_{11}^{3}$ and ${\bar{\tilde{D}}}_{22}^{3}$ versus the percent of vacancy of Si and C atoms in the SiC part of the ternary diffusion systems. It is seen that the interdiffusion coefficients increase linearly by increasing the vacancy. The average slope of ${\bar{\tilde{D}}}_{11}^{3}$ is 2.29 × 10^−11^ m^2^/s, and that of ${\bar{\tilde{D}}}_{22}^{3}$ is 2.15 × 10^−11^ m^2^/s. Furthermore, the values of ${\bar{\tilde{D}}}_{22}^{3}$  for the Si-terminated systems are higher by almost 90% than those of the C-terminated ones; however, as expected, the values of ${\bar{\tilde{D}}}_{11}^{3}$ for the C- and Si-terminated systems are almost identical.

It is worth mentioning that, based on the previous studies of other investigators, a brittle and unstable Al_4_C_3_ at the SiC/Al interface is produced at temperatures higher or equal to 923 K and up to about 1600 K (e.g., see [33]). On the other hand, the present results indicate that sufficient diffusion occurs in defective SiC/Al interface at temperatures even below 900 K. Therefore, it is expected that the formation of a diffusion zone in real defective SiC at low temperatures can help to improve the equivalent mechanical properties of the interface layer. This question will be investigated in our future studies utilizing mode I and mode II failure tests through molecular dynamics simulations.

## 3. Modeling and Simulation Method

The MD simulations are performed using the LAMMPS open-source MD software [34]. LAMMPS is a classical open-source molecular dynamics code with a focus on materials modeling. The OVITO software [35] is also used for scientific visualization and data analysis.

The embedded atom method (EAM) is one of the most popular inter-atomic potentials for face-centered cubic (fcc) metallic materials. The EAM potential obtained by Mishin et al. [36] is used to model the force between aluminum atoms. In this atomic potential, the total energy of a monoatomic system is represented by (Equation (9)) [36]:(9)$$E_{tot}=\frac{1}{2}\sum_{ij}^{}V(r_{ij})+\sum_{i}F({\bar{\rho }}_{i})$$
where *V*(*r_ij_*) is a pair potential as a function of distance *r_ij_* between atoms *i* and *j*, and *F* is the embedding energy as a function of the density ${\bar{\rho }}_{i}$ induced on atom *i* by all other atoms in the system. The density ${\bar{\rho }}_{i}$ is given by ${\bar{\rho }}_{i}=\sum_{j\neq i}\rho (r_{ij})$, where $\rho (r_{ij})$ is the atomic density function.

The bond order potential by Tersoff [37,38,39] is the most widely employed potential for SiC. The many-body Tersoff potential between atoms *i* and *j* is defined as (Equation (10)):(10)$$V_{ij}=f_{C}(r_{ij})\left[f_{R}(r_{ij})+b_{ij}f_{A}(r_{ij})\right]$$
where $f_{R}(r_{ij})$, $f_{A}(r_{ij})$, and $f_{C}(r_{ij})$ are, respectively, repulsive, attractive, and cut-off potential functions, *r_ij_* is the length of the atomic bond between atom *i* and *j*, and *b_ij_* is a function that modulates the attractive interaction.

The Lennard-Jones (LJ) and Mores potentials are the conventional potential functions for modeling interactions between ceramic and metal atoms. Dandekar and Shin [40] showed that the LJ potential cannot completely describe Al, Si, and C interactions but is rather useful in describing the adhesive interface between the components. However, the Morse potential can represent the system best and is matched by the ab initio data obtained by Zhao et al. [41]. Their proposed Morse potential function is (Equation (11)) [40]:(11)$$V=D_{0}\left[e^{-2\alpha (r-r_{0})}-2e^{-\alpha (r-r_{0})}\right]$$
where *r*, *r*_0_, *D*_0_, and α are the distance between atoms, the equilibrium bond length, the well depth of the potential, and the width of the potential, respectively. Dandekar and Shin [40] obtained the Morse potential parameters given in Table 5 for the Al–C and Al–Si interactions by curve fitting the potential function to the ab initio results. In this study, the Morse potential is used to model the interactions of atoms at the interface.

The elastic constants of fcc Al and cubic silicon carbide (3C-SiC) with lattice parameters of 4.0495 and 4.348 Å, respectively, are obtained with the aforementioned potential functions and compared with the experimental and MD simulations in Table 6. With the proposed interaction potentials, the linear elastic constants *C_ij_* are determined at zero temperature directly from the stress–strain relationship Equation (12) (e.g., see [42]):(12)$$C_{ij}=\frac{\partial \sigma_{ij}}{\partial \varepsilon_{ij}}$$
where $\sigma_{ij}$ and $\varepsilon_{ij}$ are the applied stress and strain components, respectively. The stress components are calculated from the virial stress formula. The Young modulus, Poisson ratio, shear modulus, and bulk modulus are calculated using the elastic constants as follows Equation (13):(13)$$E=\frac{\left(C_{11}-C_{12})(C_{11}+2C_{12}\right)}{C_{11}+C_{12}},\;\nu =\frac{C_{12}}{C_{11}+C_{12}},\;G=\frac{E}{2(1+\nu )},\;K=\frac{E}{3(1-2\nu )}$$

It can be seen that the present results agree very well with those obtained by the experiments and MD simulations of other investigators. Therefore, the potential functions used here can adequately model the interactions between atoms.

**Table 6 molecules-28-00744-t006:** The elastic constants, bulk modulus, Young modulus, shear modulus, and Poisson ratio obtained by the present MD simulations using the EAM and Tersoff potential functions and compared with those obtained by other MD simulations or experimental data.

Material	Method	*C*_11_ (GPa)	*C*_12_ (GPa)	*C*_44_ (GPa)	*K* (GPa)	*E* (GPa)	*G* (GPa)	*v*
Al	Present	107.03	61.06	31.05	76.38	62.67	22.99	0.363
	MD ^a^	107.21	60.60	32.88	76.14	63.44	23.31	0.361
	Experiment ^b^	107.3	60.08	28.3	75.7	63.83	23.48	0.359
3C-SiC	Present	383.78	144.41	239.75	224.20	304.81	119.68	0.273
	MD ^c^	390.1	142.7	191.0	225.1	313.6	123.7	0.268
	Experiment ^d^	390	142	256	225	314.2	124	0.267

^a^ Ref. [43]. ^b^ Ref. [44]. ^c^ Ref. [45]. ^d^ Ref. [46].

It is observed in various experimental studies that an orientation relationship exists between Al matrix and SiC reinforcements. For example, Van Drn Burg and De Hosson [47] investigated the SiC particulate-reinforced Al composite and found the orientation relationship ${\left(0001\right)}_{\alpha -\mathrm{SiC}}\|\;{\left(111\right)}_{\mathrm{Al}}$, ${[2\bar{1}\bar{1}0]}_{\alpha -\mathrm{SiC}}\|\;{\left[110\right]}_{\mathrm{Al}}$. That is, the (0001) hexagonal crystal plane of α-SiC was parallel to the (111) cubic crystal plane of Al at the interface and, in addition, the $[2\bar{1}\bar{1}0]$ direction vector of α-SiC crystal was parallel to the [110] direction vector of Al crystal. Furthermore, Li et al. [48] used a quantum chemical method to calculate the total energies of ${\left(0001\right)}_{\alpha -\mathrm{SiC}}\|\;{\left(111\right)}_{\mathrm{Al}}$, ${[2\bar{1}\bar{1}0]}_{\alpha -\mathrm{SiC}}\|\;{\left[110\right]}_{\mathrm{Al}}$ and ${[2\bar{1}\bar{1}]}_{\beta -\mathrm{SiC}}\|\;{\left(100\right)}_{\mathrm{Al}}$, ${[2\bar{1}\bar{1}]}_{\beta -\mathrm{SiC}}\|\;{\left[110\right]}_{\mathrm{Al}}$. They concluded that the bond strength between SiC and Al could be stronger than the bond between Al and Al, and that the adhesive energy in this orientation is large. Moreover, Luo et al. [49] showed that ${\left(111\right)}_{\beta -\mathrm{SiC}}\;\|\;{\left(111\right)}_{\mathrm{Al}}$, ${\left[01\bar{1}\right]}_{\beta -\mathrm{SiC}}\|\;{\left[01\bar{1}\right]}_{\mathrm{Al}}$ has a large cohesive energy consistent with a high probability of observation in the STEM experiment [50].

The α-SiC has a hexagonal crystal structure, but the β-SiC has a cubic crystal structure. In this study, one cubic (3C-SiC) and one hexagonal (6H-SiC) SiC with a higher probability of observation are modeled as one sample for each SiC crystal polytype. The orientation relationships ${\left(0001\right)}_{\alpha -\mathrm{SiC}}\|\;{\left(111\right)}_{\mathrm{Al}}$, ${[2\bar{1}\bar{1}0]}_{\alpha -\mathrm{SiC}}\|\;{\left[110\right]}_{\mathrm{Al}}$ and ${\left(111\right)}_{\beta -\mathrm{SiC}}\;\|\;{\left(111\right)}_{\mathrm{Al}}$, ${\left[01\bar{1}\right]}_{\beta -\mathrm{SiC}}\|\;{\left[01\bar{1}\right]}_{\mathrm{Al}}$ are considered for the α-SiC particulate-reinforced Al and β-SiC whisker-reinforced Al composites, respectively.

The present model consists of a dual-layer nanocomposite of SiC and Al. The lattice constants of fcc Al and cubic 3C-SiC are 4.0495 and 4.348 Å, respectively, and the lattice constants of hexagonal 6H-SiC are *a* = *b* = 3.081 Å and *c* = 15.120 Å. The initial three-dimensional MD model considered for the diffusion analysis of SiC/Al is shown in Figure 8. The grain boundaries are generated by rotating the two crystals along the appropriate rotation axis and suitable rotation angle. The initial SiC/Al interface is considered as the single crystal of Al (bottom) and single crystal of SiC (top) with an initial gap. The gap is set to the previously obtained [51] Al–C bond length (1.95 Å) for the C-terminated interfaces, the Al–Si bond length (2.41 Å) for the Si-terminated interfaces, and the average of the Al–C and Al–Si bond lengths (2.18 Å) for nonpolar SiC interfaces.

The typical dimension of the MD model is approximately 126 × 112 × 184 Å, with a total of 201,204 atoms for a defect-free diffusion system. The accurate dimensions for each model are selected considering the amount of lattice misfit between the crystal surfaces at the interface to produce Al and SiC parts with the least difference in the x- and y-directions. In addition to the intact models, to study the effect of vacancy in SiC on the diffusion of atoms, the defective models are also made by deleting randomly the Si and C atoms. In all models, two cases of C-terminated and Si-terminated configurations at the interface are analyzed to investigate the effect of atom type at the interface. The periodic boundary conditions are applied in all three directions of the samples, and a time step of 1 fs is considered throughout the simulations.

The optimization of the geometric configuration is first performed using the conjugate gradient energy minimization algorithm with a specified energy tolerance of 1 × 10^−10^ and a force tolerance of 1 × 10^−10^ eV/Å. The NVT canonical ensemble with the Nosé–Hoover thermostat at a constant temperature of 300 K is imposed on the sample for 20 ps, and then the isothermal–isobaric NPT ensemble at zero pressure and a constant temperature of 300 K is used for 30 ps to adjust the volume and relax the assembled interface system. Then, to study the diffusion of atoms, the sample is heated to a preset temperature at a heating rate of 1 K/ps. Thereafter, the temperature is maintained at the given temperature for 6.0 ns to study the interdiffusion, and the movements of Al, Si, and C atoms are recorded during this period. Finally, the sample is cooled to 300 K at a rate of 1 K/ps. Then, the structural relaxation process is performed for 20 ps under the condition of zero pressure and 300 K temperature to remove the internal residual stresses. The NPT ensemble with zero pressure is considered for all these processes. After cooling, the final sample can be used to analyze the strength of the SiC/Al and, specifically, the equivalent mechanical properties of the diffused interface in future works. To study the interdiffusion of Al in SiC, temperatures of 700, 800, 900, and 1000 K are considered for the simulations. This temperature range is taken into account since the temperature in the vacuum diffusion bonding process of SiC_p_ to Al is approximately 540 °C (813 K) [52].

## 4. Conclusions

The metal matrix composites consist of a few phases; therefore, besides the properties of each phase, the interface properties affect their overall mechanical properties. The interface between grains and phases at the atomic level is indeed a mixture of atoms of phases connected because of diffusion.

The main aim of the present study was to investigate the effects of vacancy defects in SiC, temperature, and annealing time on the interdiffusion of aluminum in silicon carbide. To this end, the diffusion systems of the Si- and C-terminated 6H-SiC/Al and 3C-SiC/Al were analyzed using molecular dynamics simulations. The self-diffusion and interdiffusion coefficients were evaluated for various diffusion systems with 10% and 20% vacancy defect contents at annealing temperatures of 700–1000 K.

The average ternary interdiffusion coefficients were obtained using the concentration profiles of atoms during diffusion, and it was seen that, as expected, the interdiffusion coefficients increase with increasing vacancy content, annealing temperature, and annealing time.

The samples after diffusion and cooling can be used in future works to estimate the effective mechanical properties of the fuzzy interface of SiC/Al and, therefore, the effective mechanical properties of SiC/Al metal matrix composites.

## Figures and Tables

**Figure 1 molecules-28-00744-f001:**
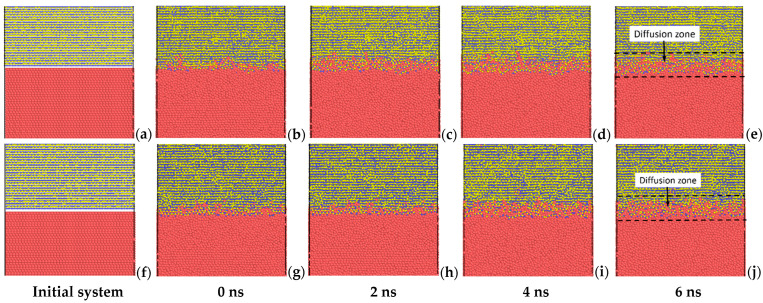
Cross-sectional views of the atomic configurations of the C-terminated (**a**–**e**) 6H-SiC/Al and (**f**–**j**) 3C-SiC/Al interfaces with 20% vacancy in SiC. The figure shows atomic structures at 300 K before relaxation and the configurations after maintaining the systems at 1000 K for 0, 2, 4, and 6 ns.

**Figure 2 molecules-28-00744-f002:**
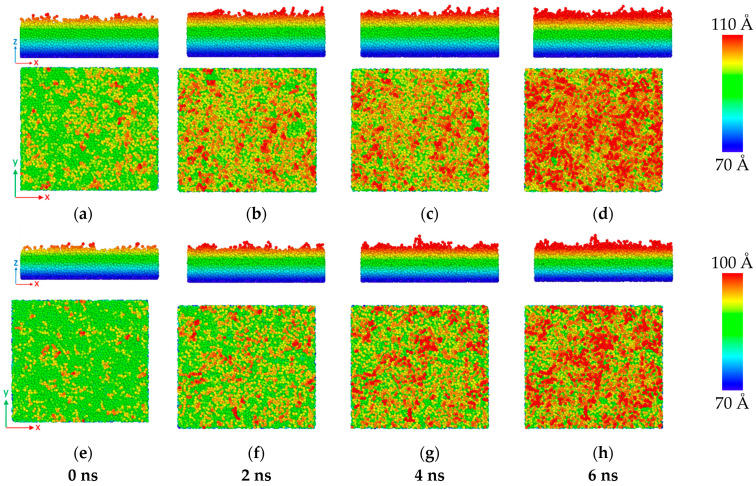
Front and top views of the atomic configurations of Al atoms in the C-terminated (**a**–**d**) 6H-SiC/Al and (**e**–**h**) 3C-SiC/Al interfaces with 20% vacancy in SiC after maintaining the systems for 0, 2, 4, and 6 ns at 1000 K. The color represents the *z*-coordinate of Al atoms near the interface.

**Figure 3 molecules-28-00744-f003:**
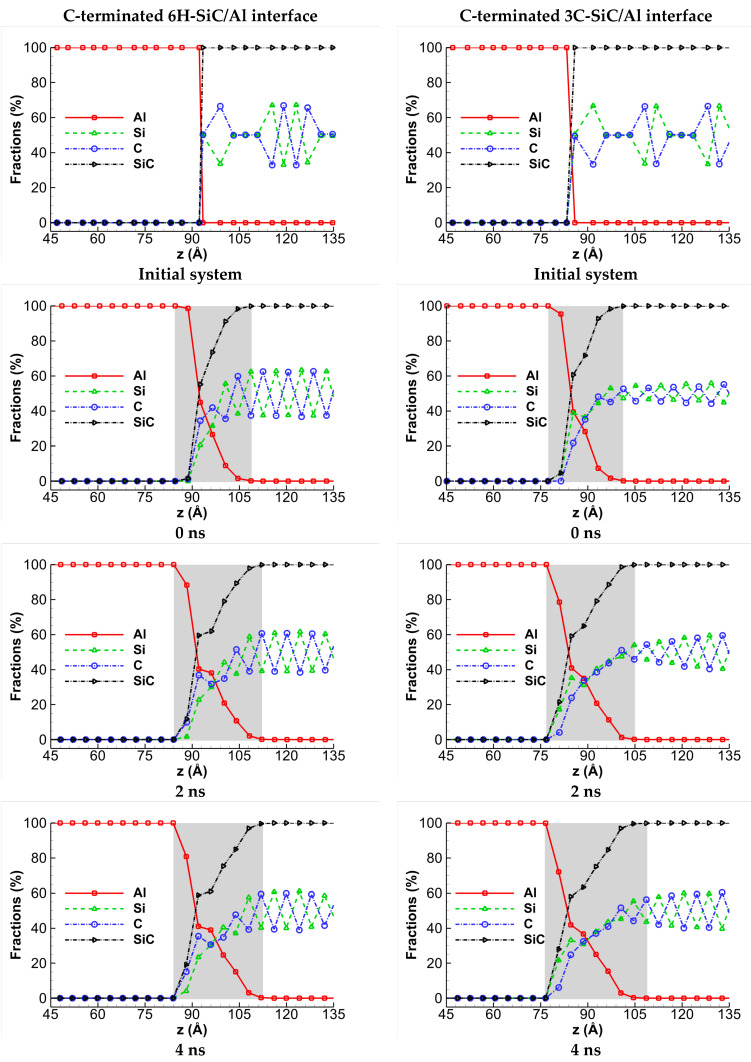
The concentration profiles of Al, Si, and C atoms along the *z*-direction during interdiffusion of the C-terminated (**a**) 6H-SiC/Al and (**b**) 3C-SiC/Al interfaces with 20% vacancy in SiC. The concentration profiles of atoms at 300 K before relaxation and after maintaining the systems at 1000 K for 0, 2, 4, and 6 ns are illustrated. The grey zone in the figures indicates the diffusion zone and its thickness measured in Å.

**Figure 4 molecules-28-00744-f004:**
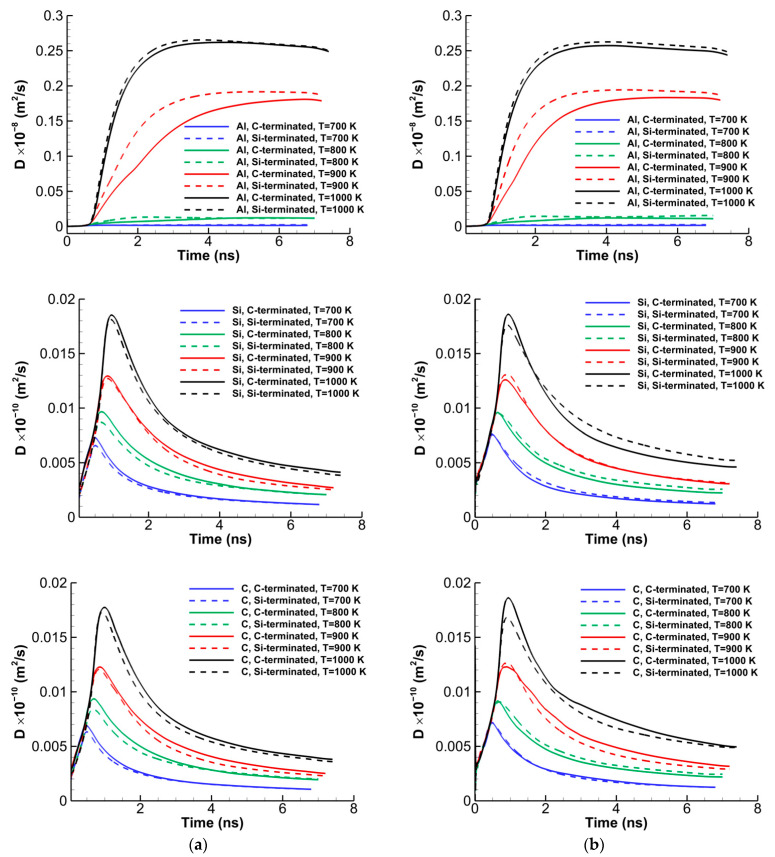
Time histories of the self-diffusion coefficients at different temperatures for different atoms of the ternary SiC/Al diffusion system with 20% vacancy in SiC in the C- and Si-terminated (**a**) 6H-SiC/Al and (**b**) 3C-SiC/Al interfaces.

**Figure 5 molecules-28-00744-f005:**
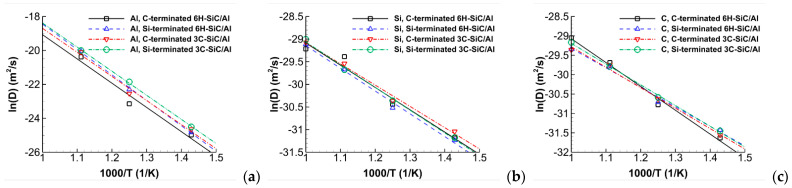
Arrhenius plots of Al, Si, and C atoms for the C- and Si-terminated 6H-SiC/Al and 3C-SiC/Al interfaces with (**a**) to (**c**) 10% and (**d**) to (**f**) 20% vacancy in SiC.

**Figure 6 molecules-28-00744-f006:**
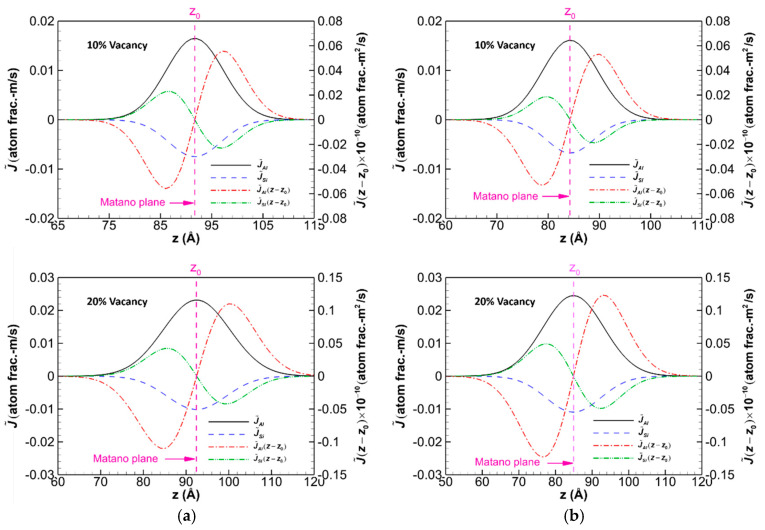
Profiles of the interdiffusion fluxes $\tilde{J}$ and $\tilde{J}(z-z_{0})$ for diffusion couples of the C-terminated (**a**) 6H-SiC/Al and (**b**) 3C-SiC/Al with 10% and 20% vacancy in SiC annealed at 1000 K for 6 ns.

**Figure 7 molecules-28-00744-f007:**
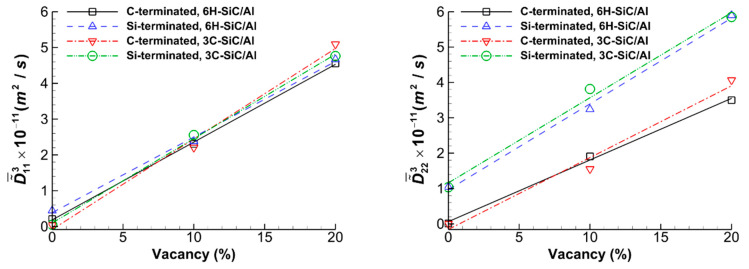
The variations in the average main interdiffusion coefficients ${\bar{\tilde{D}}}_{11}^{3}$ and ${\bar{\tilde{D}}}_{22}^{3}$ versus the percent of vacancy of Si and C atoms for the C- and Si-terminated 6H-SiC/Al and 3C-SiC/Al systems after maintaining the systems at 1000 K for 6 ns (indices: 1 = Al; 2 = Si; 3 = C).

**Figure 8 molecules-28-00744-f008:**
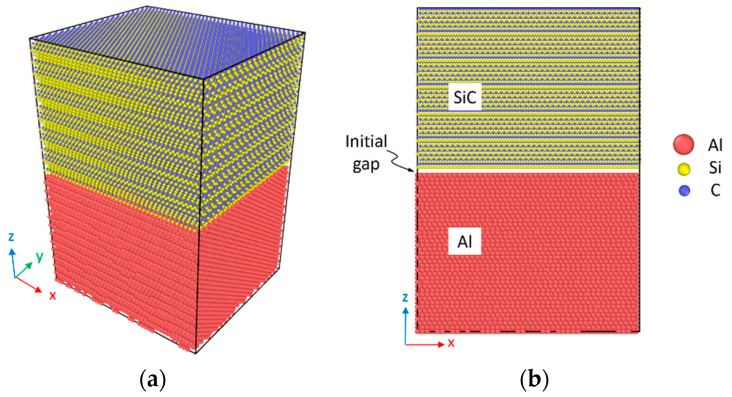
Model of the Si-terminated 6H-SiC/Al interface for the MD simulations. (**a**) Perspective view and (**b**) front view of the interdiffusion system.

**Table 1 molecules-28-00744-t001:** The self-diffusion activation energies and pre-exponential factors of Al, Si, and C for SiC/Al samples with 10% and 20% vacancy in SiC.

Vacancy in SiC	Diffusion System	Al		Si		C	
(%)		*Q* (kJ/mol)	$D_{0}$ (m^2^/s)	*Q* (kJ/mol)	$D_{0}\times 10^{-12}$ (m^2^/s)	*Q* (kJ/mol)	$D_{0}\times 10^{-12}$ (m^2^/s)
10	C-terminated 6H-SiC/Al	118.633	0.0083	40.914	32.009	51.098	110.031
	Si-terminated 6H-SiC/Al	124.508	0.0318	41.812	33.781	41.785	28.004
	C-terminated 3C-SiC/Al	118.007	0.0111	39.072	25.861	43.791	37.063
	Si-terminated 3C-SiC/Al	118.035	0.0151	41.763	36.238	44.398	43.602
20	C-terminated 6H-SiC/Al	124.601	0.0246	24.214	7.869	24.438	7.533
	Si-terminated 6H-SiC/Al	113.634	0.0055	22.762	6.130	22.886	5.813
	C-terminated 3C-SiC/Al	123.041	0.0195	25.213	9.718	26.847	12.732
	Si-terminated 3C-SiC/Al	112.447	0.0051	25.162	10.564	25.327	10.103

**Table 2 molecules-28-00744-t002:** Average values of ternary main and cross-interdiffusion coefficients on each side of the Matano plane after maintaining the system with 10% vacancy defects in SiC at the preset temperature for 6 ns (indices: 1 = Al; 2 = Si; and 3 = C).

Diffusion Couple	Temperature (K)	For Composition Range of the Bottom Side of the Matano Plane ${\bar{\tilde{D}}}_{ij}^{3}\;\times 10^{-11}(m^{2}/s)$				For Composition Range of the Top Side of the Matano Plane ${\bar{\tilde{D}}}_{ij}^{3}\;\times 10^{-11}(m^{2}/s)$			
		${\bar{\tilde{D}}}_{11}^{3}$	${\bar{\tilde{D}}}_{12}^{3}$	${\bar{\tilde{D}}}_{21}^{3}$	${\bar{\tilde{D}}}_{22}^{3}$	${\bar{\tilde{D}}}_{11}^{3}$	${\bar{\tilde{D}}}_{12}^{3}$	${\bar{\tilde{D}}}_{21}^{3}$	${\bar{\tilde{D}}}_{22}^{3}$
C-terminated 6H-SiC/Al	700	0.424	−5.4 × 10^−7^	−8.4 × 10^−9^	0.030	0.424	−1.9 × 10^−6^	−1.7 × 10^−7^	0.030
	800	0.694	−3.2 × 10^−7^	−5.0 × 10^−9^	0.057	0.694	−2.4 × 10^−6^	−4.0 × 10^−9^	0.057
	900	1.194	−2.9 × 10^−6^	−1.6 × 10^−7^	0.783	1.194	3.9 × 10^−6^	3.4 × 10^−8^	0.783
	1000	2.309	4.4 × 10^−6^	7.7 × 10^−7^	1.909	2.309	−2.7 × 10^−6^	3.4 × 10^−7^	1.909
Si-terminated 6H-SiC/Al	700	0.421	1.1 × 10^−7^	−6.9 × 10^−7^	1.238	0.421	1.8 × 10^−7^	−1.8 × 10^−7^	1.238
	800	0.805	1.7 × 10^−7^	1.5 × 10^−8^	1.422	0.805	−9.2 × 10^−7^	−5.9 × 10^−7^	1.422
	900	1.487	2.2 × 10^−6^	1.8 × 10^−7^	2.088	1.487	2.7 × 10^−6^	3.8 × 10^−6^	2.088
	1000	2.381	−4.4 × 10^−7^	−7.7 × 10^−7^	3.239	2.381	−4.6 × 10^−7^	−8.3 × 10^−7^	3.239
C-terminated 3C-SiC/Al	700	0.480	4.3 × 10^−8^	2.6 × 10^−8^	0.241	0.480	−1.7 × 10^−6^	2.2 × 10^−7^	0.241
	800	0.761	2.8 × 10^−6^	−6.6 × 10^−7^	0.675	0.762	1.4 × 10^−6^	5.2 × 10^−7^	0.675
	900	1.313	3.6 × 10^−6^	−2.1 × 10^−6^	1.198	1.313	7.9 × 10^−6^	1.4 × 10^−8^	1.198
	1000	2.204	−1.1 × 10^−6^	−7.5 × 10^−8^	1.552	2.204	−3.1 × 10^−6^	8.0 × 10^−8^	1.552
Si-terminated 3C-SiC/Al	700	0.423	1.2 × 10^−6^	5.7 × 10^−8^	0.305	0.423	−2.4 × 10^−6^	3.1 × 10^−7^	0.305
	800	0.718	−1.6 × 10^−6^	4.7 × 10^−7^	0.608	0.718	−2.2 × 10^−6^	5.8 × 10^−7^	0.608
	900	1.211	2.1 × 10^−6^	−1.0 × 10^−7^	1.411	1.211	3.3 × 10^−6^	−2.9 × 10^−7^	1.411
	1000	2.555	3.0 × 10^−7^	−5.4 × 10^−8^	3.814	2.555	−1.2 × 10^−7^	−3.8 × 10^−7^	3.814

**Table 3 molecules-28-00744-t003:** Average values of ternary main and cross-interdiffusion coefficients on each side of the Matano plane after maintaining the system with 20% vacancy defects in SiC at the preset temperature for 6 ns (indices: 1 = Al; 2 = Si; and 3 = C).

Diffusion Couple	Temperature (K)	For Composition Range of the Bottom Side of the Matano Plane ${\bar{\tilde{D}}}_{ij}^{3}\;\times 10^{-11}(m^{2}/s)$				For Composition Range of the Top Side of the Matano Plane ${\bar{\tilde{D}}}_{ij}^{3}\;\times 10^{-11}(m^{2}/s)$			
		${\bar{\tilde{D}}}_{11}^{3}$	${\bar{\tilde{D}}}_{12}^{3}$	${\bar{\tilde{D}}}_{21}^{3}$	${\bar{\tilde{D}}}_{22}^{3}$	${\bar{\tilde{D}}}_{11}^{3}$	${\bar{\tilde{D}}}_{12}^{3}$	${\bar{\tilde{D}}}_{21}^{3}$	${\bar{\tilde{D}}}_{22}^{3}$
C-terminated 6H-SiC/Al	700	1.657	5.9 × 10^−6^	2.2 × 10^−7^	1.292	1.657	−1.4 × 10^−5^	7.6 × 10^−7^	1.292
	800	2.420	3.1 × 10^−6^	5.1 × 10^−7^	1.701	2.420	−1.8 × 10^−6^	4.3 × 10^−7^	1.701
	900	3.528	−1.5 × 10^−6^	−1.9 × 10^−7^	2.562	3.528	−2.7 × 10^−6^	1.2 × 10^−7^	2.562
	1000	4.554	−1.6 × 10^−6^	−6.1 × 10^−7^	3.498	4.554	−3.9 × 10^−6^	4.1 × 10^−7^	3.498
Si-terminated 6H-SiC/Al	700	2.019	4.4 × 10^−7^	−2.8 × 10^−7^	2.583	2.019	2.8 × 10^−6^	−2.8 × 10^−7^	2.583
	800	2.792	8.9 × 10^−7^	1.6 × 10^−7^	3.487	2.792	6.0 × 10^−7^	4.9 × 10^−7^	3.487
	900	3.782	−2.1 × 10^−6^	−2.7 × 10^−7^	4.852	3.782	−2.4 × 10^−6^	−8.5 × 10^−7^	4.852
	1000	4.668	9.0 × 10^−7^	−1.6 × 10^−5^	5.897	4.668	9.3 × 10^−6^	−8.2 × 10^−6^	5.897
C-terminated 3C-SiC/Al	700	1.805	3.2 × 10^−7^	−3.8 × 10^−7^	1.338	1.805	−8.3 × 10^−7^	2.2 × 10^−7^	1.338
	800	2.791	5.3 × 10^−7^	−1.4 × 10^−7^	1.903	2.791	3.8 × 10^−7^	−1.1 × 10^−7^	1.903
	900	3.732	−3.2 × 10^−7^	1.2 × 10^−7^	2.784	3.732	−8.8 × 10^−7^	1.3 × 10^−7^	2.784
	1000	5.092	−7.6 × 10^−5^	7.4 × 10^−8^	4.066	5.092	−4.8 × 10^−4^	−1.0 × 10^−6^	4.066
Si-terminated 3C-SiC/Al	700	1.405	−6.4 × 10^−7^	2.4 × 10^−7^	2.102	1.405	−1.2 × 10^−6^	3.4 × 10^−7^	2.102
	800	2.270	1.2 × 10^−6^	1.4 × 10^−6^	2.992	2.270	7.6 × 10^−7^	1.6 × 10^−6^	2.992
	900	3.423	7.6 × 10^−7^	−2.9 × 10^−6^	4.099	3.423	2.7 × 10^−6^	−1.6 × 10^−6^	4.099
	1000	4.762	2.3 × 10^−4^	−5.3 × 10^−4^	5.851	4.762	2.3 × 10^−4^	−5.4 × 10^−4^	5.851

**Table 4 molecules-28-00744-t004:** Average values of main and cross-interdiffusion coefficients of the defect-free ternary SiC/Al systems on each side of the Matano plane after maintaining the systems at 1000 K for 6 ns (indices: 1 = Al; 2 = Si; and 3 = C).

Diffusion Couple	For Composition Range of the Bottom Side of the Matano Plane ${\bar{\tilde{D}}}_{ij}^{3}\;\times 10^{-11}(m^{2}/s)$				For Composition Range of the Top Side of the Matano Plane ${\bar{\tilde{D}}}_{ij}^{3}\;\times 10^{-11}(m^{2}/s)$			
	${\bar{\tilde{D}}}_{11}^{3}$	${\bar{\tilde{D}}}_{12}^{3}$	${\bar{\tilde{D}}}_{21}^{3}$	${\bar{\tilde{D}}}_{22}^{3}$	${\bar{\tilde{D}}}_{11}^{3}$	${\bar{\tilde{D}}}_{12}^{3}$	${\bar{\tilde{D}}}_{21}^{3}$	${\bar{\tilde{D}}}_{22}^{3}$
C-terminated 6H-SiC/Al	0.209	2.9 × 10^−8^	9.1 × 10^−8^	0.011	0.209	4.4 × 10^−8^	7.0 × 10^−8^	0.011
Si-terminated 6H-SiC/Al	0.444	3.5 × 10^−8^	7.6 × 10^−9^	1.042	0.444	2.9 × 10^−7^	−2.2 × 10^−6^	1.042
C-terminated 3C-SiC/Al	0.037	−5.2 × 10^−7^	5.9 × 10^−9^	0.002	0.037	−2.7 × 10^−7^	2.2 × 10^−8^	0.002
Si-terminated 3C-SiC/Al	0.048	3.1 × 10^−8^	1.7 × 10^−8^	1.035	0.048	1.0 × 10^−8^	−8.2 × 10^−8^	1.035

**Table 5 molecules-28-00744-t005:** The Morse potential parameters to model Al–Si and A–C interactions [41].

System	Parameters	Morse Potential
Al–Si	*D*_0_ (eV)	0.4824
	α (1/Å)	1.322
	*r*_0_ (Å)	2.92
Al–C	*D*_0_ (eV)	0.4691
	α (1/Å)	1.738
	*r*_0_ (Å)	2.246

## Data Availability

Not applicable.

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
