# Peer review of "Effect of Vacancy Defect Content on the Interdiffusion of Cubic and Hexagonal SiC/Al Interfaces: A Molecular Dynamics Study"

_molecules, 2023, doi:10.3390/molecules28020744_

Round 1

Reviewer 1 Report

The authors have done interesting works and some representative data were obtained. I know little about the simulation process, so I will only suggest a few aspects related to composites:

1. As you mentioned in the introduction, SiC/Al interfacial reaction to form Al4C3 would degrade the mechanical properties. So what is the significance of the studied intra-interface diffusion behavior of SiC/Al? 

2. All consequences were obtained under the situation that the diffusion time was set to 6 ns, this differed greatly from the actual diffusion time, and how to use these results to relate to the actual experimental situation.

3. The annealing temperature was set to 700-1000k,  How is this considered?

Author Response

Dear Dr. Cirillo and Prof. Long,

Thank you very much for the valuable and useful comments of your respected reviewers on our manuscript. Even though we have done our best to modify the manuscript based on your esteemed reviewers' comments, we welcome any further comments if the respected reviewers wish to raise them.

Below please find our point-by-point answers to the reviewers' questions/concerns. It should be noted please that our responses to the first respected reviewer are written in RED and to the second respected reviewer in BLUE.

Sincerely yours,

Masoud Tahani

Reviewer 1

Open Review

English language and style

( ) English very difficult to understand/incomprehensible
( ) Extensive editing of English language and style required
( ) Moderate English changes required
( ) English language and style are fine/minor spell check required
(x) I don't feel qualified to judge about the English language and style

Yes

Can be improved

Must be improved

Not applicable

Does the introduction provide sufficient background and include all relevant references?

( )

(x)

( )

( )

Are all the cited references relevant to the research?

(x)

( )

( )

( )

Is the research design appropriate?

(x)

( )

( )

( )

Are the methods adequately described?

(x)

( )

( )

( )

Are the results clearly presented?

(x)

( )

( )

( )

Are the conclusions supported by the results?

(x)

( )

( )

( )

Comments and Suggestions for Authors

The authors have done interesting works and some representative data were obtained. I know little about the simulation process, so I will only suggest a few aspects related to composites:

Response: We appreciate the positive impression of the respected reviewer and have done our best to carefully address all the concerns raised by the respected reviewer.

  1. As you mentioned in the introduction, SiC/Al interfacial reaction to form Al4C3 would degrade the mechanical properties. So what is the significance of the studied intra-interface diffusion behavior of SiC/Al?

Response: Thanks for your interesting question and concern. The previous studies showed that the major problem encountered during the fabrication of SiC/Al metal-matrix composites is the reactivity of SiC with molten aluminum at higher processing temperatures (e.g., see [52]). The silicon carbide is attacked by pure aluminum at temperatures higher or equal to 923 K and up to about 1600 K, according to the chemical reaction [52]:

The current study found that Al diffuses in the defective SiC in temperatures even below 900 K and, according to the previous studies, in these temperatures, the above reaction to form aluminum carbide does not occur. However, Al diffuses in intact SiC at higher temperatures.

On the other hand, it is well known that in the small-scale production of SiC, a variety of defects occur. The carbon and silicon vacancy defects have been observed as the most important point defects in SiC [4-7]. Therefore, we expect that the formation of a diffusion zone in real defective SiC at low temperatures can help to improve the equivalent mechanical properties of the interface layer. We will investigate this question in our future studies via mode I and mode II failure tests utilizing molecular dynamics to obtain cohesive law for the diffusion zone. In other words, we will compare the traction-separation law of the interface region for samples before diffusion at room temperature and samples that were annealed at a high temperature and cooled to room temperature.

According to the concern raised by the respected reviewer, we decided to add the following explanations to the revised manuscript on page 17, lines 422-429:

It is worth mentioning that, based on previous studies of other investigators, a brittle and unstable Al4C3 at the SiC/Al interface is produced at temperatures higher or equal to 923 K and up to about 1600 K (e.g., see [52]). On the other hand, the present results indicated that sufficient diffusion occurs in defective SiC/Al interface at temperatures even below 900 K. Therefore, it is expected that the formation of a diffusion zone in real defective SiC at low temperatures can help to improve the equivalent mechanical properties of the interface layer. This question will be investigated in our future studies utilizing mode I and mode II failure tests through molecular dynamics simulations.

  1. All consequences were obtained under the situation that the diffusion time was set to 6 ns, this differed greatly from the actual diffusion time, and how to use these results to relate to the actual experimental situation.

Response: Thank you very much for your concern. It is worth mentioning that the present work was a numerical attempt to evaluate the interdiffusion of Al in SiC and obtain the composition of atoms in the diffused zone and thickness of this region as functions of annealing temperature and time. To ensure numerical stability, the time steps in an MD simulation must be short, typically only a few femtoseconds each. Nevertheless, most physical processes occur in much longer timescales, several orders of magnitude over our current research capabilities. To this end, it is not possible to set a long time such as experimental studies in MD analysis because of huge simulation costs. On the other hand, it is well known that the magnitude of the interdiffusion coefficient depends on the annealing time [R1]. Therefore, our findings obtained by the MD simulations provide a useful guide for applications in forming SiC and Al bonding. The simulated results indicated that the thickness of SiC/Al diffusion layer increased with increasing diffusion time, and interdiffusion results in disordered or amorphization in the diffusion zone, which are generally consistent and in line with the experimental observations in diffusion couples.

Similar problem exists during tensile loading in MD simulations and it is not possible to compare directly the stress-strain curves of the MD simulations with experiments. In MD simulations of tensile test, the simulation volume is small and strain rate is very high because of restriction of simulation costs. Recently, Koyanagi et al. [R2] studied the tensile strength of a polymer material using molecular dynamics and compared their results with experiments. They found that the tensile strength obtained by MD simulation is almost always higher than the experimental value. They determined strength as a function of the simulation volume based on Weibull statistics and then the relationship was extrapolated to a much higher number of molecules, which was equivalent to a real specimen. Also, the relationship between the tensile strength and strain rate was determined and it was extrapolated to match the strain rate in actual experiments. Consequently, a predicted strength was close to the experimental result.

According to the explanations given above, similar methods, such as those proposed by Koyanagi et al. [R2], for matching the results of the molecular dynamics with those of experiments will most likely be presented in the future.  

References

[R1] Olaye, O.; Ojo, O.A. Time variation of concentration-dependent interdiffusion coefficient obtained by numerical simulation analysis. Materialia 2021, 16, 101056, doi:https://doi.org/10.1016/j.mtla.2021.101056.

[R2] Koyanagi, J.; Takase, N.; Mori, K.; Sakai, T. Molecular dynamics simulation for the quantitative prediction of experimental tensile strength of a polymer material. Composites Part C: Open Access 2020, 2, 100041, doi:https://doi.org/10.1016/j.jcomc.2020.100041.

  1. The annealing temperature was set to 700-1000k, How is this considered?

Response: Thanks for your question and concern. It is worth mentioning that the vacuum diffusion bonding process is commonly used for joining discontinuously reinforced metal-matrix composites such as SiCp-reinforced aluminum (e.g., see [41]). The diffusion bonding temperature for joining SiC to Al is about 540°C or 813 K [41]. Hence, in this study, we considered the annealing temperature in the range of 700-1000 K. We believed that Al diffuses in the defective SiC in lower temperature conditions and the present findings confirmed this prediction. However, as expected, the thickness of the diffusion zone increase with increasing the annealing temperature and time.

To clarify the reason for considering this temperature range, we decided to add the following explanation on page 7, lines 250-252:

This temperature range is taken into account since the temperature in the vacuum diffusion bonding process of SiCp to Al is approximately 540°C (813 K) [40].”

Reviewer 2 Report

Review

In the current manuscript, the diffusional exchange of atoms across the interface of a double-layer silicon carbide (SiC) and aluminum (Al) is simulated using molecular dynamics simulation. In particular, the effect of defect and vacancy in SiC is studied where randomly removal of atoms are considered to account for the defects. The SiC is deemed as a reinforcement in Al matrix composites and two crystalline structure (i.e. cubic and hexagonal) are considered to represent whisker and particulate nano reinforcement. The contribution addresses an interesting issue which is important in mechanical evaluation of nanocomposites. However, the manuscript requires some modifications. Therefore, I would recommend the current work to be considered for “Molecules”, providing that in a revised minor revision the Authors address the following issues and improve the manuscript.

Remarks:

·       The Authors have mentioned that the equivalent mechanical properties of the diffused interface will be checked in future works. However, the exact procedure to calculate the elastic properties of Al and SiC can be easily followed to evaluate the mechanical parameters of a double-layer Al-SiC before and after annealing to show the effect of interdiffusion.

·       The representation used to show the relative crystal orientation of nanoparticle and matrix should be described the first time it appears in the text (for instance in line 172, (0001)α-SiC ||(111) Al and [2110] α-SiC || [110] Al).

·       Fig. 4 is of interest; however, the caption is not clear and needs to be revised. It is recommended to write the time for each of the figure right below them to clearly show the interdiffusion progress versus time.

·       The literature review is mainly focused on SiC-Al nanocomposites. It is recommended that it will be improved to provide a more comprehensive overview and highlight the importance of interface\interphase on the mechanical properties of nanocomposites. Some recommended references are: DOI: 10.1016/j.pmatsci.2006.09.003, DOI: 10.1140/epjp/s13360-021-01819-9, DOI: 10.1016/j.physb.2019.08.013. Besides, since the interdiffusion calculations is the principle topic of the work, the literature review should include at least a brief survey on the interdiffusion methods in nanocomposites. Only referring the readers to a reference is not proper. 

·       In the title the word “molecular” should initiate with capital letters.

·       Inconsistent font is used through the manuscript.

·       Regarding details of MD simulations, the potentials are clearly described, however, no information on the method to extract elastic constants is provided. For instance the Authors can refer to DOI: 10.1016/j.physb.2019.08.013 for a description of the procedure to find elastic parameters.

Author Response

Dear Dr. Cirillo and Prof. Long,

Thank you very much for the valuable and useful comments of your respected reviewers on our manuscript. Even though we have done our best to modify the manuscript based on your esteemed reviewers' comments, we welcome any further comments if the respected reviewers wish to raise them.

Below please find our point-by-point answers to the reviewers' questions/concerns. It should be noted please that our responses to the first respected reviewer are written in RED and to the second respected reviewer in BLUE.

Sincerely yours,

Masoud Tahani

Reviewer 2

Open Review

English language and style

( ) English very difficult to understand/incomprehensible
( ) Extensive editing of English language and style required
( ) Moderate English changes required
(x) English language and style are fine/minor spell check required
( ) I don't feel qualified to judge about the English language and style

Yes

Can be improved

Must be improved

Not applicable

Does the introduction provide sufficient background and include all relevant references?

( )

( )

(x)

( )

Are all the cited references relevant to the research?

(x)

( )

( )

( )

Is the research design appropriate?

( )

(x)

( )

( )

Are the methods adequately described?

( )

( )

(x)

( )

Are the results clearly presented?

( )

( )

(x)

( )

Are the conclusions supported by the results?

( )

(x)

( )

( )

Comments and Suggestions for Authors

Review

In the current manuscript, the diffusional exchange of atoms across the interface of a double-layer silicon carbide (SiC) and aluminum (Al) is simulated using molecular dynamics simulation. In particular, the effect of defect and vacancy in SiC is studied where randomly removal of atoms are considered to account for the defects. The SiC is deemed as a reinforcement in Al matrix composites and two crystalline structure (i.e. cubic and hexagonal) are considered to represent whisker and particulate nano reinforcement. The contribution addresses an interesting issue which is important in mechanical evaluation of nanocomposites. However, the manuscript requires some modifications. Therefore, I would recommend the current work to be considered for “Molecules”, providing that in a revised minor revision the Authors address the following issues and improve the manuscript.

Response: We appreciate the positive impression of the respected reviewer and have done our best to carefully address all the concerns raised by the respected reviewer.

Remarks:

  • The Authors have mentioned that the equivalent mechanical properties of the diffused interface will be checked in future works. However, the exact procedure to calculate the elastic properties of Al and SiC can be easily followed to evaluate the mechanical parameters of a double-layer Al-SiC before and after annealing to show the effect of interdiffusion.

Response: It is well known that the nature of interface has a strong influence over the properties of the metal-matrix composites (MMCs). Most of the mechanical and physical properties of the MMCs are dependent on the interfacial behavior. The interface plays a crucial role in transferring the load efficiently from the matrix to the reinforcement. Hence, it is essential to evaluate the interface bonding of SiC/Al. To this end, the first step in this multi-scale study is the modeling of the interdiffusion of atoms at the interface at different annealing temperatures and times. The next step will be estimating the equivalent mechanical properties or cohesive law of the fuzzy interface by mode I and mode II fracture tests using the molecular dynamics method. Finally, the last step will be large-scale modeling of SiC/Al composites using the mechanical properties of Al, SiC, and cohesive zone model of the interface to obtain overall stiffness and strength of this composite. Hence, this manuscript is the first step in determining the overall mechanical properties of SiC/Al composite.

In other words, we do not intend to obtain equivalent elastic properties of the interface, however, we are interested to obtain an equivalent cohesive zone model with the traction-separation law for the interface layer and study the effect of diffusion on the this law. The traction-separation law will be determined by doing mode I and mode II failure tests through molecular dynamics. 

To clarify the meaning of the equivalent mechanical properties of the interface, we decided to add the following short explanation on pages 2-3, lines 91-94:

A cohesive zone model with the traction-separation law is traditionally utilized to characterize the interface. The traction-separation relationships are determined by performing mode I and mode II failure tests for a ductile-brittle system.”

  • The representation used to show the relative crystal orientation of nanoparticle and matrix should be described the first time it appears in the text (for instance in line 172, (0001)α-SiC ||(111) Al and [2110] α-SiC || [110] Al).

Response: The authors wish to thank the respected reviewer for his/her suggestion for completing this manuscript. We added the description concerning the representation of the relative crystal orientation on page 5, lines 196-199 as follows:

That is, (0001) hexagonal crystal plane of a-SiC was parallel to (111) cubic crystal plane of Al at interface and, in addition,  direction vector of a-SiC crystal was parallel to [110] direction vector of Al crystal.

  • Fig. 4 is of interest; however, the caption is not clear and needs to be revised. It is recommended to write the time for each of the figure right below them to clearly show the interdiffusion progress versus time.

Response: Thanks for your justified comment. According to your suggestion that helps reading the manuscript easier, the diffusion holding times of the system at 1000 K was added below each figure. Moreover, the caption of this figure was revised as:

- Page 10, lines 296-300: Figure 4. The concentration profiles of Al, Si, and C atoms along the z-direction during interdiffusion of the C-terminated (a) 6H-SiC/Al and (b) 3C-SiC/Al interfaces with 20% vacancy in SiC. The concentration profiles of atoms at 300 K before relaxation and after maintaining the systems at 1000 K for 0, 2, 4, and 6 ns are illustrated. The grey zone in the figures indicates the diffusion zone and its thickness.”

  • The literature review is mainly focused on SiC-Al nanocomposites. It is recommended that it will be improved to provide a more comprehensive overview and highlight the importance of interface\interphase on the mechanical properties of nanocomposites. Some recommended references are: DOI: 10.1016/j.pmatsci.2006.09.003, DOI: 10.1140/epjp/s13360-021-01819-9, DOI: 10.1016/j.physb.2019.08.013. Besides, since the interdiffusion calculations is the principle topic of the work, the literature review should include at least a brief survey on the interdiffusion methods in nanocomposites. Only referring the readers to a reference is not proper. 

Response: Authors wish to express their thanks to the respected reviewer for his/her contribution to the successful completion of this manuscript. According to your suggestion and your introduced references, we decided to add the following explanations on page 3, lines 95-102:

Moya et al. [10] investigated the challenges of ceramic/metal micro/nanocomposites in the new technologies. They reviewed the exotic effects of metal particles embedded into matrix ceramics due to the dissimilar properties of the components, percolation laws, and the nature of the interfaces. The interested reader will find sufficient references in this review article to cover the literature in more depth concerning several aspects of ceramic/metal interfaces including the role of interface in the fracture toughness and wear resistance of the composite and the wettability issue of dissimilar ceramic and metal materials to reach an appropriate adherence.”

  • In the title the word “molecular” should initiate with capital letters.

Response: It was corrected.

  • Inconsistent font is used through the manuscript.

Response: The authors thank the respectful reviewer for the careful reading of the manuscript. We polished the whole text of our manuscript carefully such that it would become free of any typos and inconsistent fonts.

  • Regarding details of MD simulations, the potentials are clearly described, however, no information on the method to extract elastic constants is provided. For instance the Authors can refer to DOI: 10.1016/j.physb.2019.08.013 for a description of the procedure to find elastic parameters.

Response: The main objective of this manuscript was the interdiffusion of a ternary system. To keep the paper short, no explanation was provided concerning how the engineering constants are calculated. However, according to your justified suggestion, the following explanation was added on page 5, lines 177-181, of the revised manuscript:

With the proposed interaction potentials, the linear elastic constants Cij are determined at zero temperature directly from the stress-strain relationship (e.g., see [31]):

(4)

where  and  are the applied stress and strain components, respectively. The stress components are calculated from the virial stress formula.”
